# Mapping cell-to-tissue graphs across human placenta histology whole slide images using deep learning with HAPPY

Claudia Vanea [1,2] ✉, Jelisaveta Džigurski [3], Valentina Rukins [3], Omri Dodi[4], Siim Siigur[5], Liis Salumäe[5], Karen Meir[6], W. Tony Parks [7], Drorith Hochner-Celnikier[4], Abigail Fraser[8,9], Hagit Hochner [10], Triin Laisk [3], Linda M. Ernst[11,12], Cecilia M. Lindgren[2,13,14,15,16] & Christoffer Nellåker [1,2,16] ✉

Accurate placenta pathology assessment is essential for managing maternal and newborn health, but the placenta's heterogeneity and temporal variability pose challenges for histology analysis. To address this issue, we developed the 'Histology Analysis Pipeline.PY' (HAPPY), a deep learning hierarchical method for quantifying the variability of cells and micro-anatomical tissue structures across placenta histology whole slide images. HAPPY differs from patch-based features or segmentation approaches by following an interpretable biological hierarchy, representing cells and cellular communities within tissues at a single-cell resolution across whole slide images. We present a set of quantitative metrics from healthy term placentas as a baseline for future assessments of placenta health and we show how these metrics deviate in placentas with clinically significant placental infarction. HAPPY's cell and tissue predictions closely replicate those from independent clinical experts and placental biology literature.

Accurate placenta pathology assessment is essential for clinical management of mother and newborn health[1–3]. Placental pathology informs immediate clinical treatment, predicts recurrence risk in subsequent pregnancies and long-term maternal and child adverse outcomes, and explains underlying causes of pregnancy loss[3,4]. Ongoing placental pathology research can determine new biomarkers of disease, such as the recently identified SARS-CoV-2 placentitis[5,6], and improve our understanding of the biological mechanisms of healthy development and disease processes[3].

Despite its importance, the placenta remains a relatively understudied and poorly understood organ[1,7–16]. It is the first organ formed by the developing foetus and is heterogeneous[17–19] and rapidly evolving[2,7,20,21]. The placenta's high spatial and temporal variability presents a challenge for robust, reproducible histological analysis and detection of pathological changes. The difficulty of this task is reflected in low interobserver reliability among pathologists for many diagnostically-relevant placental features (i.e. gestational age[22], villus maturity[23], maternal vascular malperfusion[24–26]). Placenta histology

[1]Nuffield Department of Women's & Reproductive Health, University of Oxford, Oxford, UK. [2]Big Data Institute, Li Ka Shing Centre for Health Information and Discovery, University of Oxford, Oxford, UK. [3]Institute of Genomics, University of Tartu, Tartu, Estonia. [4]Faculty of Medicine, Hadassah Hebrew University Medical Center, Jerusalem, Israel. [5]Department of Pathology, Tartu University Hospital, Tartu, Estonia. [6]Department of Pathology, Hadassah Hebrew University Medical Center, Jerusalem, Israel. [7]Department of Laboratory Medicine & Pathobiology, University of Toronto, Toronto, Canada. [8]Population Health Sciences, Bristol Medical School, University of Bristol, Bristol, UK. [9]MRC Integrative Epidemiology Unit at the University of Bristol, Bristol, UK. [10]Braun School of Public Health, Hebrew University of Jerusalem, Jerusalem, Israel. [11]Department of Pathology and Laboratory Medicine, NorthShore University HealthSystem, Chicago, USA. [12]Department of Pathology, University of Chicago Pritzker School of Medicine, Chicago, USA. [13]Centre for Human Genetics, Nuffield Department, University of Oxford, Oxford, UK. [14]Broad Institute of Harvard and MIT, Cambridge, MA, USA. [15]Nuffield Department of Population Health Health, University of Oxford, Oxford, UK. [16]These authors jointly supervised this work: Cecilia M. Lindgren, Christoffer Nellåker,
✉e-mail: claudia.vanea@wrh.ox.ac.uk; christoffer.nellaker@bdi.ox.ac.uk

slides commonly contain upwards of a million cells comprising tens of thousands of tissue microstructures. High-throughput, quantitative and objective metrics of placental biology are therefore valuable for placental investigations in both clinical and research settings.

Digital pathology has the potential to provide these high-throughput metrics, allowing for automated processing of histology slides at scale. In recent years, deep learning is becoming a gold standard for digital pathology, with success in cancer survival analysis[27–29] and tumour micro-environment modelling[30–34]. However, there has been little application of these approaches to placenta histology[19,35–37], where reliable quantification of micro-anatomical structures across slides is vital to determining key features of health assessment such as placental maturity[18,20,21,38].

Established deep learning approaches for quantifying micro-anatomical structures in histology have been largely based on tissue segmentation[39–43] or prediction at the patch level[34,44–46]. Tissue segmentation typically requires a large amount of precise manually curated annotations to perform well[40–43] and while patch-based approaches only require patch-level labels, prediction resolution is limited to patch size. Additionally, as both approaches operate within fixed patches, they are unable to utilise contextual information external to a patch and are vulnerable to changes in patch construction[45].

In recent years, the representation of cells as a spatial graph[28,47–57] has emerged as a promising way to discover new cellular phenotypes[49,53,57], make slide-level predictions[28,48,51,56], and hierarchically cluster cells into tissue microstructures[31,50,54,55]. However, for the most common imaging modality, Hematoxylin and Eosin (H&E) stained histology, these have only been applied for patch-level[31,50,52] and slide-level prediction[28,48,51,56] or on graphs restricted to fixed subregions[50,54]. Here we present a pipeline for analysing tissue microstructures at the single-cell level by building whole slide spatial cell graphs across H&E histology with dynamic sampling of graph regions and learnt cellular community aggregation. This pipeline leverages the relatively large cell sample size within a slide to overcome uncertainty and variance in any one classification and to achieve robust tissue microstructure classification.

## Results

### Automated quantification of healthy variability in placenta histology

We present HAPPY (Histology Analysis Pipeline.PY), a method for quantifying cells and micro-anatomical tissue structures across Hematoxylin and Eosin (H&E) stained placenta histology whole slide images (WSIs). Our approach is inspired by the biological hierarchy of the organ, from locating all nuclei across a WSI, to classifying their cell types, to identifying the tissue microstructures that those cells comprise. HAPPY can facilitate large-scale morphometric studies of placenta histology, a currently manual, expert-requiring, labour-intensive task.

HAPPY classifies all cells in a placenta parenchyma WSI into one of 11 cell types and 9 tissue microstructure categories. We train and validate nuclei localisation and cell classification models on 11,755 nuclei and 13,842 cells and evaluate on a held-out test set of 2754 nuclei and 2743 cells. We train and validate a graph neural network tissue classification model on 468,869 nodes and evaluate on a held-out portion of 179,095 test nodes across microstructures. We compare the tissue classification model's performance with the labels and Cohen's kappa agreement scores of four practising perinatal pathologists over 180 tissues and we use these to validate the ground truth training annotations and model performance. We show how in 30 WSIs from 30 healthy term placentas the predicted proportions of cells, tissues, and composition of tissues correspond to expectations from our current understanding of placental biology. We present these findings as new quantitative metrics for placental health. Finally, we present a pilot

case study of clinically significant placental infarction in 12 WSIs from eight term placentas and show how HAPPY identifies biologically-relevant differences in cell and tissue microstructures compared to the healthy group.

HAPPY is structured as a deep learning pipeline in three stages: i) an object detection model for nuclei localisation, ii) an image classification model for cell classification, and iii) a graph neural network (GNN) for tissue classification (Fig. 1). From the nuclei locations and cell predictions, we construct a spatial cell graph across the WSI. To model the interactions of cells within tissue microstructures, we use inductive message-passing node classification across the constructed cell graph. This hierarchical, message-passing approach induces our models to use the constituent cellular information to understand tissue microstructures. By reducing the WSI dimensionality to cell representations we are able to model tissue microstructures at the single cell resolution. The trained tissue model, based on an inductive GNN, is robust to input graphs of any shape or size allowing for application across other WSIs. We apply this methodology to automated whole slide-scale quantification of cells and tissue microstructures in placenta histology (Fig. 2).

The HAPPY codebase, training data, and trained models for placenta histology are available at (https://github.com/Nellaker-group/happy). The codebase supports the most commonly used WSI scanner formats[58–60], has additional utilities for creating datasets and visualising outputs, and we provide tutorials for training and inference workflows. HAPPY is presented here applied to the placenta, but by design the codebase can be directly applied to other organ histology, given organ-specific training data. To show that this is valid in principle, we have additionally conducted a preliminary investigation of our nuclei localisation and cell classification models across WSIs of a placenta membrane roll, umbilical cord, a second-trimester placenta with chorioamnionitis, and also across WSIs of other organs in the GTEx dataset (Supplementary Fig. 1).

### Evaluation of model performance

We evaluate the deep learning model from each stage of the pipeline on respective unseen held-out test sets (Table 1). The nuclei localisation model achieves a 0.884 F1 score across 2754 nuclei within 38 images, comparable to F1 scores reported by other state-of-the-art nuclei detection models trained for other organs (HoVer-Net achieves an F1 score of 0.756 on the CoNSeP dataset[61] and 0.800 on the Pan-Nuke dataset[62], SONNET[63] achieves an F1 score of 0.855 on the MoNuSAC dataset[64]). The cell classification model, evaluated across 2743 cells for 11 placental cell types, achieves an overall accuracy of 84.29% and a top-2 accuracy of 94.90%, with a 0.9773 macro-averaged Receiver Operating Characteristic Area Under Curve (ROC AUC). We show (Fig. 3a) that most misclassifications are within closely related cell differentiation pathways. See Supplementary Fig. 2 for visualisations of predictions across WSIs of healthy-term placentas.

The graph neural network tissue classification model, evaluated across 149,425 cell graph nodes for 9 tissue types, achieves an overall accuracy of 68.34% and a top-2 and top-3 accuracy of 91.14% and 97.10%, with a 0.8868 macro-averaged ROC AUC. We show (Fig. 3b) that misclassifications of tissues fall primarily within developmentally similar microstructures. Misclassifications of villus types are typically confused with other villus types which correspond to similarities in villus growth and branching morphology[20,21]. For example, mature intermediate villi, from which terminal villi grow, are mistaken for terminal villi 37% of the time. Likewise, anchoring villi, which are a subcategory of stem villi and have the same cellular composition, are mislabelled as stem villi 21% of the time. Avascular villi, which are commonly associated with the presence of fibrin[65–67], are confounded with fibrin 21% of the time. Given the high top-2 accuracy, model misclassifications likely correspond to noise inherent in the biological domain. Many tissue microstructure types are not discrete categories

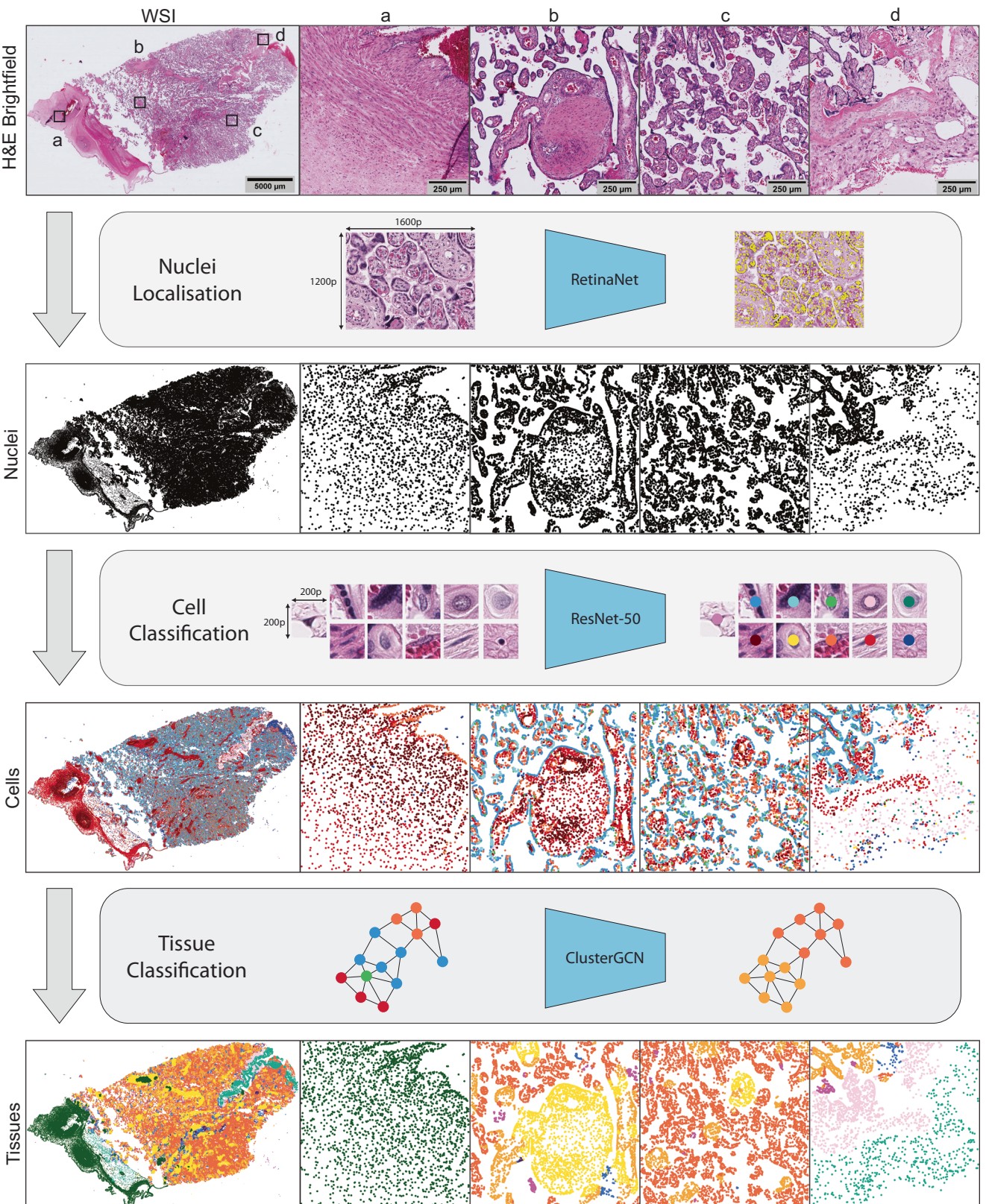

**Fig. 1 | HAPPY workflow.** A hematoxylin and eosin (H&E) stained whole slide image (WSI) is first sectioned into overlapping 1600 × 1200 (177.44 × 133.08 μm) pixel images and passed to an object detection RetinaNet model which identifies the nuclei in these images. 200 × 200 (22.18 × 22.18 μm) pixel images centred on each nucleus are classified into one of 11 cell types by a ResNet-50 model. The 64-dimension embeddings from the cell classifier and their nucleus coordinates are used to build a cell graph across the whole slide image. The cell graph is input into a ClusterGCN graph neural network which classifies the tissue microstructure to which each cell belongs. Images a-d show characteristic tissue regions of the WSI: **a** chorionic plate, **b** stem and distal villi, **c** distal villi, **d** basal plate and anchoring villi.

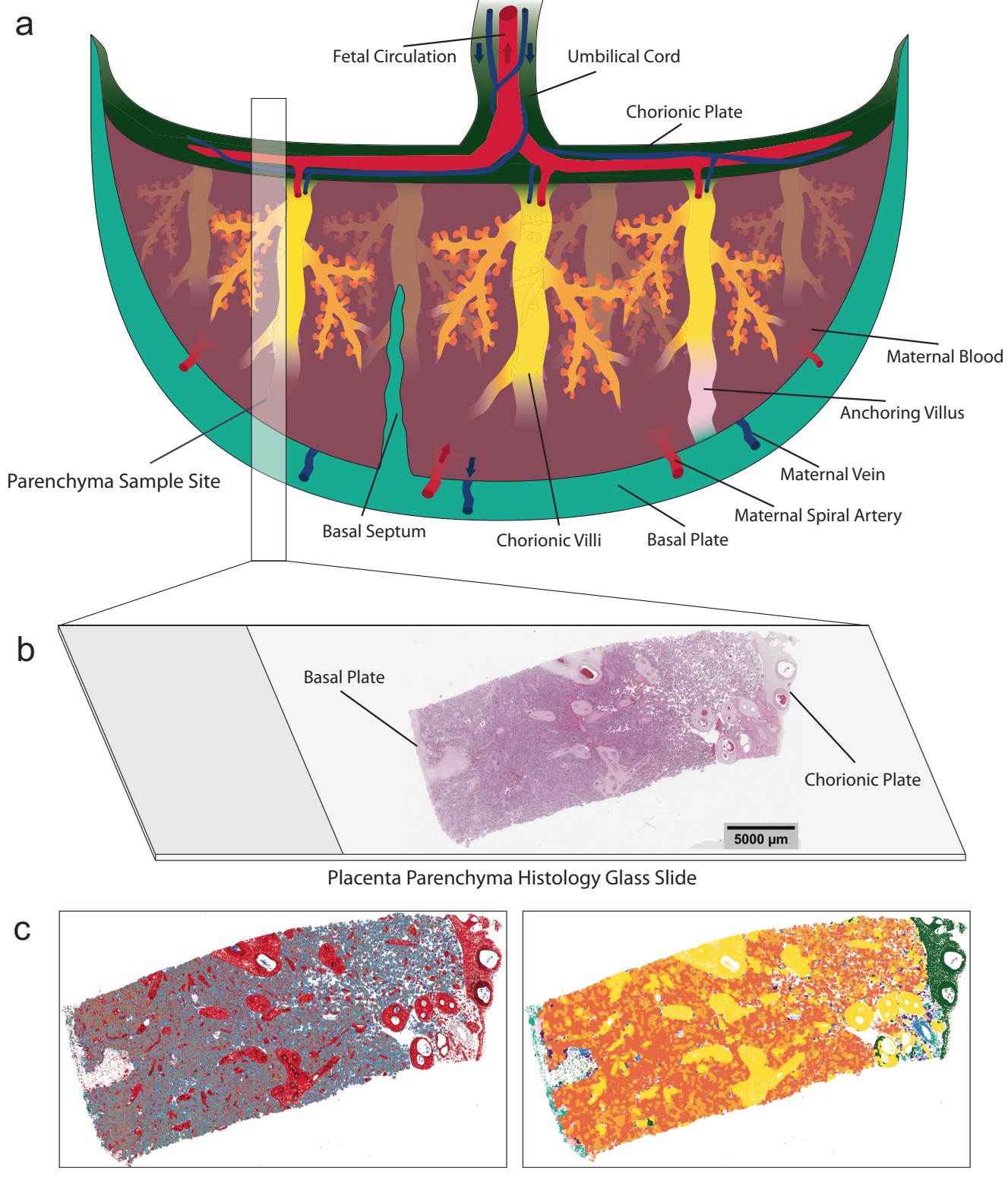

**Fig. 2 | Schematic diagram of a healthy term placenta with a histology image and model whole slide cell and tissue predictions. a** Healthy term placenta schematic showing macrostructures including the basal plate, chorionic plate and branching chorionic villi (not drawn to-scale). The fetal vasculature extends from the umbilical cord, through the chorionic plate and into the chorionic villi for maternal/fetal diffusive exchange. The intervillous space is supplied with maternal blood by the maternal spiral arteries and deoxygenated blood and waste is removed by the maternal veins. An example parenchyma slide sample site is shown by the semi-transparent box. **b** Resulting histology section presented on a glass slide. **c** Whole slide cell and tissue microstructure predictions from HAPPY.

but rather sit on a biological continuum (see Fig. 2 and further details below). A cell which appears at the blurred boundary of two villus types will have been manually labelled into one category, but the model is not necessarily incorrect to classify it as the other. Despite these challenges, the signal identified from the noise is useful for biological and clinical characterisation (see sections 'Quantitative metrics for placental health' and 'A Case Study of Placental Infarction'). Supplementary Fig. 3 contains visualisations of predictions across WSIs of healthy-term placentas.

### Comparison to perinatal pathologists

To better understand the relative difficulty of reliably identifying different placental tissue microstructures and to further validate our approach, we compare the agreement of four expert perinatal pathologists, K.M, S.S, W.T.P, L.M.E, in 180 images. Across all tissue microstructure types, pathologists have a moderate agreement score of 0.55 kappa, with low-moderate kappa values for mature

intermediate villi (0.468), villus sprouts (0.371), avascular villi (0.154), and anchoring villi (0.051). Pathologists disagree with their majority-voted label (Fig. 4a) at least 50% of the time for anchoring villi and avascular villi, highlighting the difficulty of identifying these structures.

Taking the pathologists' majority label as the gold standard, for all 180 images we compare against the manual annotations created by C.V. (used as ground truth for training the tissue model). The resulting kappa value of 0.61 indicates a slightly better agreement than inter-pathologists agreement. For 7/9 tissue microstructures, the pathologists' labels match the annotations >50% of the time (Fig. 4b). There is a strong match for terminal villi (78%), avascular villi (80%), chorionic plate tissue (90%) and basal plate tissue (99%). Of the two structures with <50% label match, mature intermediate villi (41%) and anchoring villi (27%), these were among the structures with the lowest inter-pathologist agreement as described above.

We contrast model PR-AUC (Precision Recall Area Under Curve) values against pathologist agreement scores for each tissue microstructure type (Fig. 4c). The PR-AUC values have a strong positive correlation ($R^2 = 0.821$) with the Cohen's kappa between the pathologists, suggesting the model's predictions are on par with perinatal pathologists for this task. Additionally, the pathologists' label disagreement (Fig. 4a) shows similar patterns to model confusion (Fig. 3b).

Pathologist disagreement in this task is not unexpected as specific, structure-by-structure tissue classification is not part of pathology investigations, partly because this is not humanly feasible at scale. Nonetheless, the gestalt or organised whole, i.e. tissue type and

**Table 1 | Summary of performance on unseen test data for each deep learning stage**

|  | F1 |  |  |
| --- | --- | --- | --- |
| **Nuclei Detection** | **0.884** |  |  |
|  | Accuracy | Top-2 Accuracy | ROC AUC |
| Cell Classification | 84.29% | 94.90% | 0.977 |
| Tissue Classification | 68.34% | 91.42% | 0.888 |

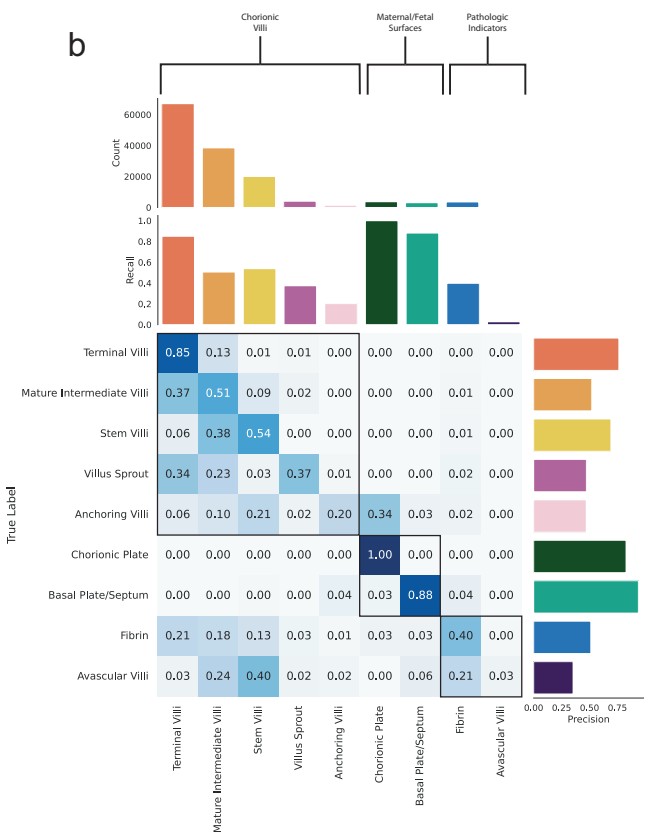

**Fig. 3 | Cell and tissue classifier performance on unseen test data. a** Confusion matrix and precision and recall values of cell classifier predictions. Cell types are clustered into categories (trophoblast, mesenchymal-derived, extravillus) and ordered by their counts. Categories are highlighted by the topmost brackets and squares within the confusion matrix. **b** Confusion matrix and precision and recall values of tissue classifier predictions. Confusion matrix values are represented as a proportion of predictions relative to the number of samples per tissue type. Tissue types are clustered into categories (chorionic villi, maternal/fetal surfaces, pathologic indicators) and ordered by their counts. Categories are highlighted by the topmost brackets and squares within the confusion matrix. Source data are provided as a Source Data file.

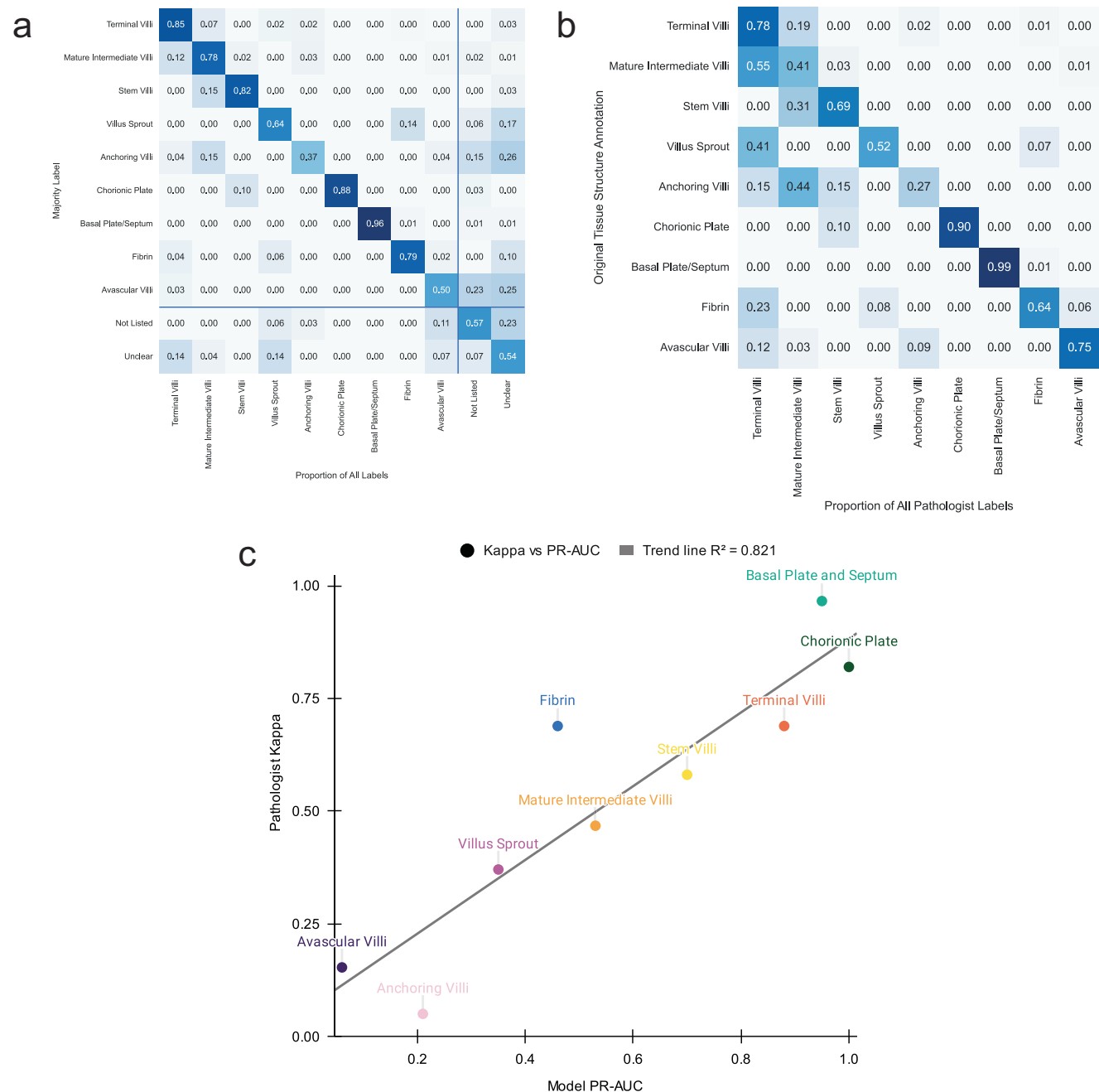

**Fig. 4 | Agreement and model confusion across tissue types. a** Inter-pathologist confusion matrix showing for each image with a majority pathologist label (y-axis) how much variance there was against that majority (x-axis). Lines indicate the additional options for pathologists to say the tissue was unclear or not listed. **b** Confusion matrix which shows the proportion of matching pathologists' labels with ground truth annotations. **c** Model performance Precision Recall Area Under Curve (PR-AUC) scores plotted against pathologist Kappa values by tissue type showing a strong positive correlation ($R^2 = 0.821$). Source data are provided as a Source Data file.

morphology assessment in aggregate, is a key part of pathology reporting and disease prediction[38]. Specific tissue classification performance comparable to human experts shows models can accurately quantify placenta biology in a way that is likely relevant to pathology detection. These results highlight the potential of large-scale deep-learning methods to identify abnormalities in placental micro-structures which are too subtle to be recognised by routine light microscopy examination.

### Quantitative metrics for placental health

We show how HAPPY can provide new cellular and tissue micro-structure quantitative metrics for assessing placental health, and we

compare these outputs to expectations from our current under-standing of placental biology. In Fig. 5a, we present the distribution and variability of predicted cells as a proportion of all cells within a WSI, from 30 parenchyma WSIs of healthy term placentas. We describe how these predictions reflect the expected internal anatomy of a healthy term placenta. The high proportion of syncytio-trophoblasts (>40%) relative to villus stromal cells matches the expected large surface area to volume ratio of an effective villus tree system optimised for diffusive exchange[21]. The <1% proportion of undifferentiated mesenchymal cells and low proportion of cyto-trophoblasts (4%–13%) are characteristic of the late maturation stage of the placenta samples[20,21,38]. The low proportion of

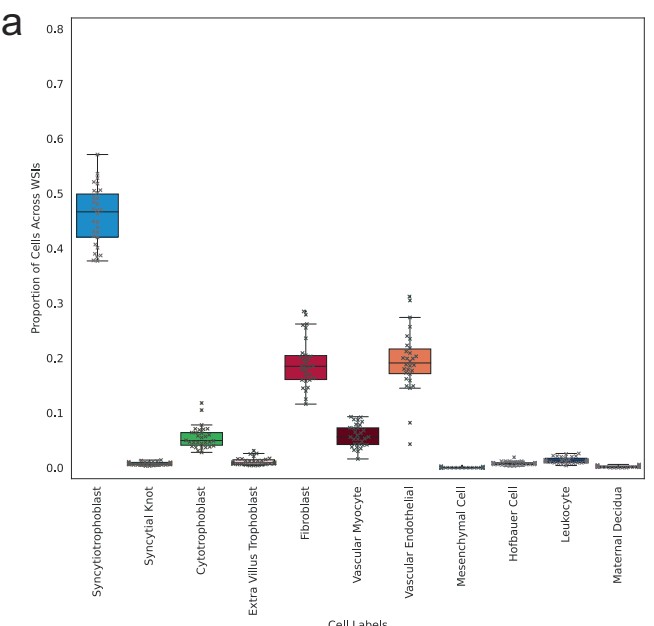

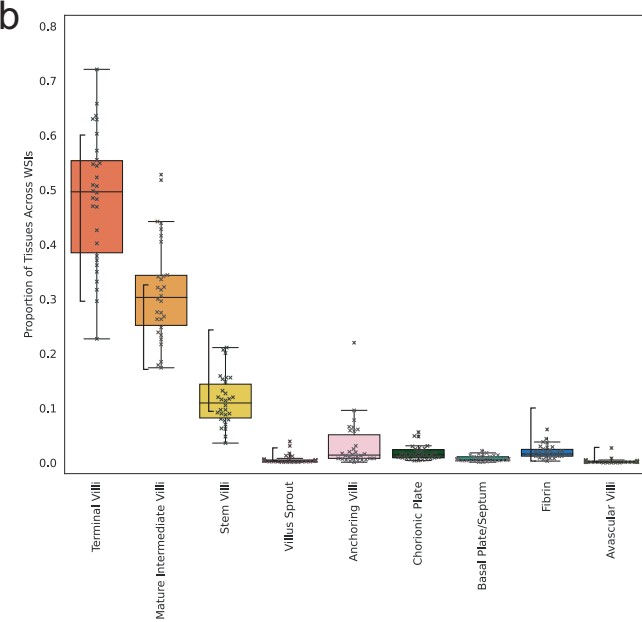

**Fig. 5 | Predicted cell and tissue type proportions across 30 whole slide images (WSI) from healthy term placentas. a** Cell proportions across WSIs. **b** Tissue proportions across WSIs with vertical braces showing the expected ranges reported in the literature for healthy term placentas[13,21,38,66,69–71]. Box centre line represents the median and whiskers are drawn up to 1.5 times the interquartile range. Each WSI datapoint is shown by a cross marker. Source data are provided as a Source Data file.

leucocytes (2%–4%) are below clinical thresholds for pathological relevance[68].

Next, we analyse the distribution and variability of predicted tissue microstructures as a proportion of all tissues within a WSI, from 30 parenchyma WSIs of healthy term placentas. We observe that the proportion of villus microstructures predominantly fall within the ranges reported in the literature for healthy-term placentas (Fig. 5b)[2,13,21,66,69,70]. In the majority of slides, terminal villi are the most common villus type comprising an expected 30–60% of the slide's tissue structures. Likewise, mature intermediate villi and stem villi are the 2nd and 3rd most common tissue structures, predominantly falling within the expected ranges of 17–32% and 9–25%, respectively. Excessive regions of fibrin or avascular villi are indicators of pathologic processes[65,71] and the proportions of fibrin and avascular villi are <10% and <2.5%, respectively, which are below the clinical thresholds for pathological relevance[13,65,71,72].

We predict the mean cellular compositions for the five chorionic villus tissue types[21] across 30 WSIs (Fig. 6a). The predicted cellular proportions match current descriptions and schematics reported in placenta literature[20,21] (Supplementary Table 1). The hierarchical nature of the pipeline reflects the inherent hierarchical relationship between cells and tissues and provides a biologically meaningful way to interpret model predictions. As villus types develop from one another in a tree-like structure (Fig. 6b), their cellular proportions shift along the tree in a continuum and model predictions recapitulate this continuum. For example, the terminal villi, which form the tips of the villus tree and are the primary sites for maternal/fetal diffusive exchange, are characterised by their >50% capillary stromal volume[20,21]. As such, their cellular composition contains the largest proportion of vascular endothelial cells. Conversely, the stem villi, which form the trunk of the villus tree and support the villus structure[20,21], contain a large proportion of structural fibroblasts and vascular myocytes.

**A case study of placental infarction**
Placental infarction is a lesion of the placental parenchyma whereby a region or regions of villi undergo ischaemic coagulative necrosis[73].

Caused by a disruption of the maternal circulation within the placental space, it is a key reported finding in pathological investigation as a marker of maternal vascular malperfusion[71,73,74]. At term, when clinically significant[71], it is associated with maternal hypertensive disease, abruption, and fetal growth restriction[75,76].

We compare cell and tissue microstructure predictions between 30 WSIs of healthy term placentas against 12 WSIs of term placentas with clinically significant placental infarction. See Supplementary Figs. 4 and 5 for visualisations of cell and tissue microstructure predictions across these slides. Given the biological changes caused by placental infarction[73], at the cellular level we would expect to see fewer cells found in healthy distal villi such as syncytiotrophoblast, cytotrophoblast, fibroblast and vascular endothelial cells and more extravillus trophoblasts and leucocytes. In terms of tissue microstructures, we would expect there to be fewer distal villi such as the terminal and mature intermediate villi and, in their place, there should be a larger proportion of fibrin and villi without vasculature (avascular villi). As we do not adjust for the age of the infarction and as younger infarctions will exhibit less nuclear degeneration, we expect these changes to sit along a continuum. We test the significance of these cell and tissue microstructure differences independently between our two groups using two-sided Welch's t-test.

Contrasting the proportion of cells across our samples (Fig. 7), we see that syncytiotrophoblast ($p = 0.001$), fibroblast ($p = 0.002$), and vascular endothelial cells (p < 0.001) are nominally significantly fewer in placentas with infarction and extravillus trophoblast cells ($p = 0.001$) and leucocytes ($p = 0.002$) are significantly higher. In terms of the tissue microstructures, there are fewer terminal villi (p < 0.001) and mature intermediate villi ($p = 0.03$) and more fibrin ($p = 0.002$) and avascular villi ($p = 0.001$). Additionally, these proportions of fibrin and avascular villi surpass the healthy expected ranges reported in the literature for 10/12 and 8/12 WSIs with placental infarction, respectively. Similarly, 9/12 and 4/12 WSIs with placental infarction have proportions of terminal villi and mature intermediate villi below expected ranges for healthy term placentas.

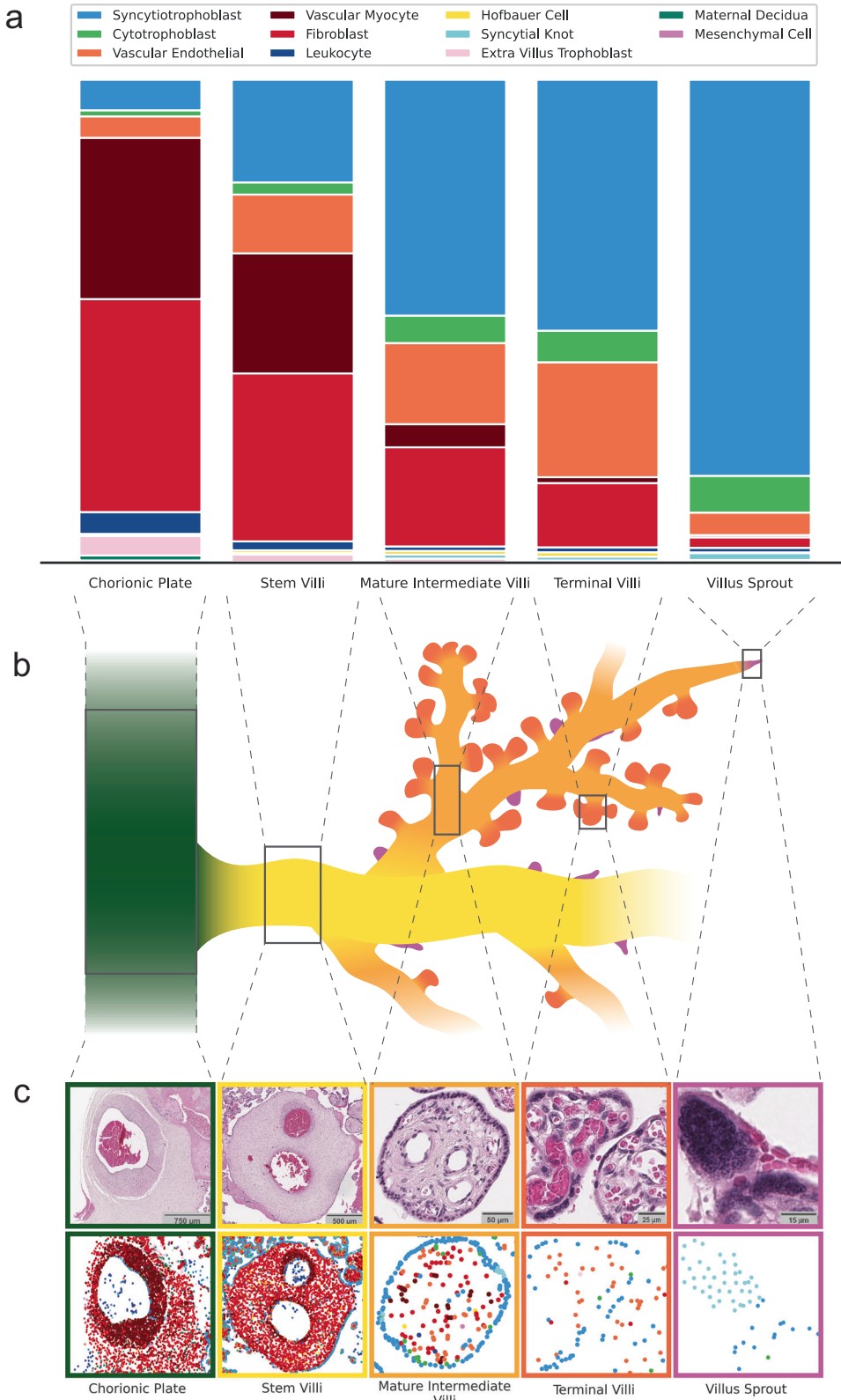

**Fig. 6 | Proportions of predicted cell types within predicted term chorionic villus tissue microstructures and their corresponding locations in a villus tree schematic. a** The mean proportion of predicted cell types within each predicted term chorionic villus tissue microstructure for 30 whole slide images. **b** A villus tree schematic showing how each villus structure relates to and grows from other villus structures. **c** Examples from histology for each villus structure with the top row containing the raw histology and the bottom row containing cell predictions. Source data are provided as a Source Data file.

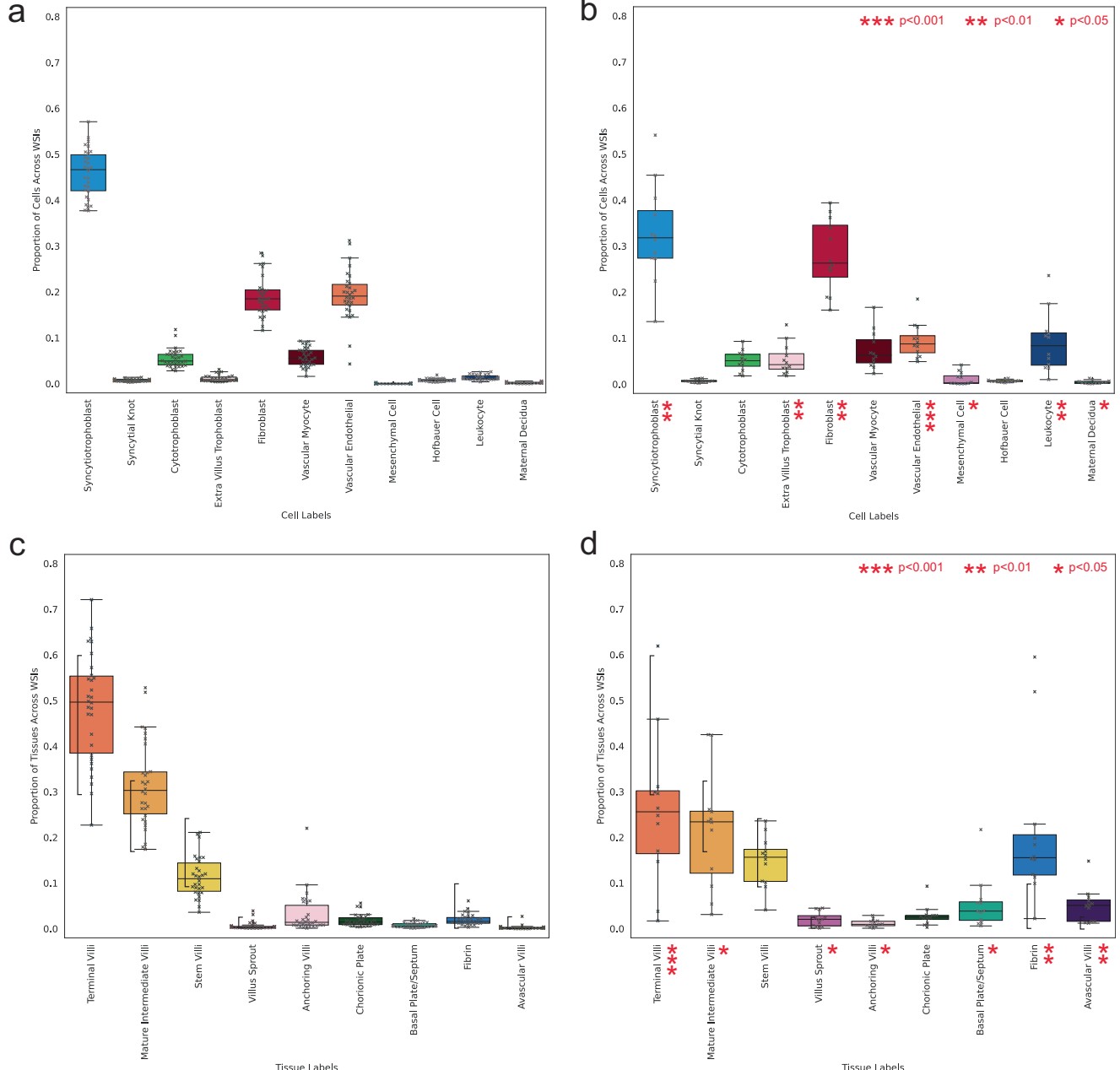

**Fig. 7 | Predicted cell and tissue microstructure proportions across 30 whole slide images (WSI) of healthy term placentas and 12 WSIs of placentas with placental infarction.** **a** Cell proportions across healthy term placentas, **b** cell proportions across term placentas with placental infarction, **c** tissue proportions across healthy term placentas, **d** tissue proportions across term placentas with placental infarction. Box centre line represents the median and whiskers are drawn up to 1.5 times the interquartile range. Each WSI datapoint is shown by a cross marker. Expected healthy ranges for tissue microstructures, as reported in the literature, are shown by black vertical bars. Nominal significant differences in cell and tissue structures between the two groups are calculated using two-sided

Welch's t-test and shown by red asterisks. Statistically significant cell proportion differences are seen for syncytiotrophoblasts ($p = 0.001$), fibroblasts ($p = 0.002$), vascular endothelial cells ($p < 0.001$), extravillus trophoblasts ($p = 0.001$), leucocytes ($p = 0.002$), mesenchymal cells ($p = 0.02$) and maternal decidual cells ($p = 0.04$). Statistically significant tissue proportion differences are seen for terminal villi ($p < 0.001$), mature intermediate villi ($p = 0.03$), villus sprouts ($p = 0.01$), anchoring villi ($p = 0.02$), basal plate/septum ($p = 0.02$), fibrin ($p = 0.002$) and avascular villi ($p = 0.001$). Source data are provided as a Source Data file.

Given that placental infarction will result in fewer total nuclei across a slide, we additionally compare the number of predicted cell and tissue microstructure counts per mm² area of tissue on the slide (Fig. 8). We estimate this area by splitting the slide into non-overlapping patches and aggregating the area of patches containing at least one nucleus prediction. There is significantly fewer total nuclei density ($p < 0.001$) in the WSIs with placental infarction. We observe similar nominally significant results for expected cell and

tissue type densities across the slides. After Bonferroni multiple testing correction, syncytiotrophoblast ($p = 0.002$), extravillus trophoblast ($p = 0.01$), vascular endothelial cells ($p < 0.001$), leucocytes ($p = 0.006$) and total density ($p = 0.003$) remain significant. Likewise, for the tissue microstructure types, terminal villi ($p = 0.002$), mature intermediate villi ($p = 0.009$), anchoring villi ($p = 0.03$), fibrin ($p = 0.007$) and avascular villi ($p = 0.003$) remain significant.

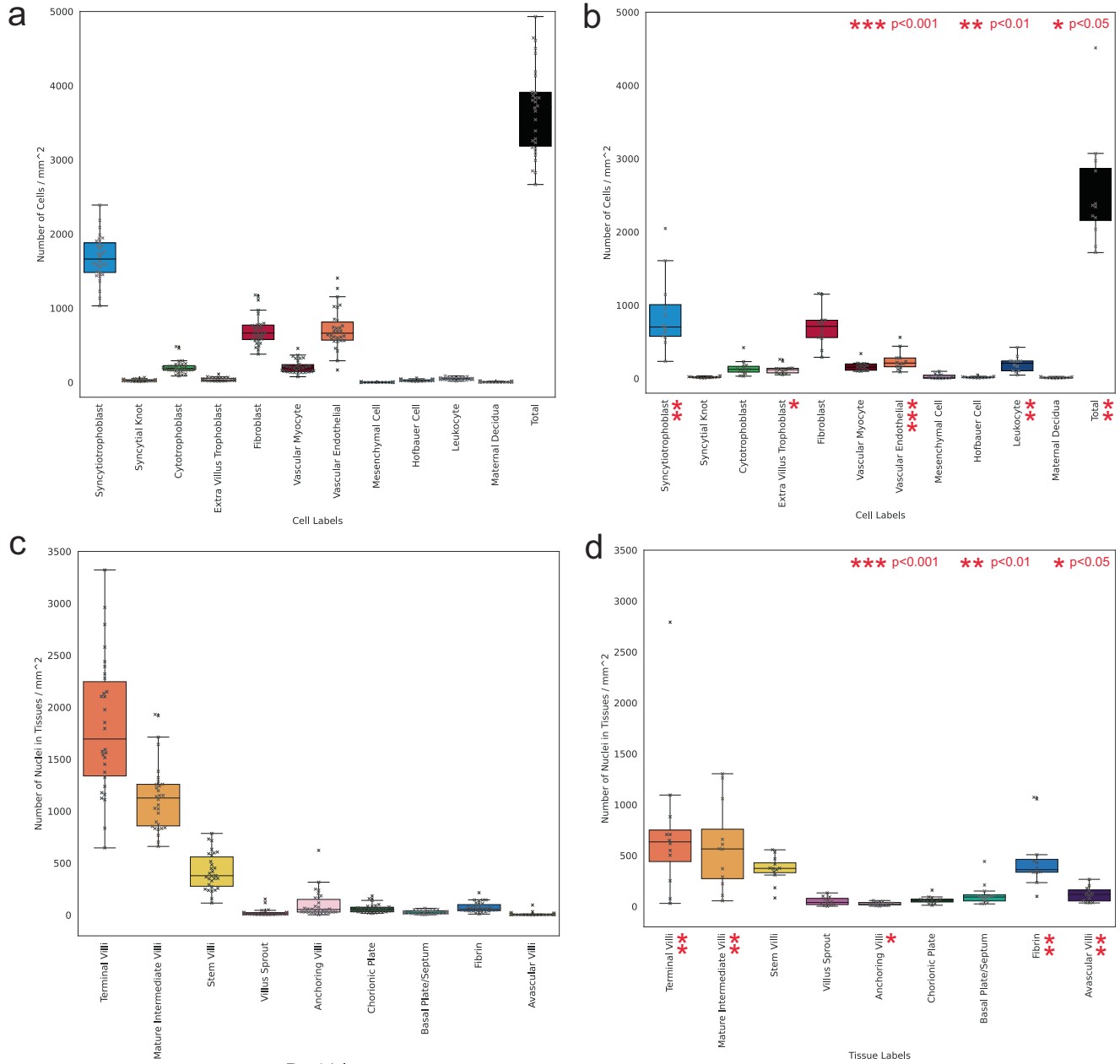

**Fig. 8 | Predicted cell and tissue microstructure density per area mm² across 30 whole slide images (WSI) of healthy term placentas and 12 WSIs of placentas with placental infarction. a** Cell density across healthy term placentas including total nuclei, **b** cell density across term placentas with placental infarction including total nuclei, **c** tissue node density across healthy term placentas, **d** tissue node density across term placentas with placental infarction. Box centre line represents the median and whiskers are drawn up to 1.5 times the interquartile range. Each WSI datapoint is shown by a cross marker. Bonferroni adjusted significant differences in cell and tissue structures between the two groups are calculated using two-sided Welch's t-test and shown by red asterisks. Adjusted statistically significant cell density differences are seen for syncytiotrophoblasts ($p = 0.002$), extravillus trophoblasts ($p = 0.01$), vascular endothelial cells ($p < 0.001$), leukocytes ($p = 0.006$) and total density ($p = 0.003$). Adjusted statistically significant tissue density differences are seen for terminal villi ($p = 0.002$), mature intermediate villi ($p = 0.009$), anchoring villi ($p = 0.03$), fibrin ($p = 0.007$) and avascular villi ($p = 0.003$). Source data are provided as a Source Data file.

## Generalisability and domain shift

H&E stain intensity can vary between institutes for several reasons (stain concentrations and ratios, soak duration, slice thickness, different slide scanners) and can vary within an institute across days and operators[77,78]. For models to robustly generalise to future data, they must be invariant to these stain differences. We explore how data augmentation, including our custom H&E stain augmentation, impacts the stain invariance of the nuclei detection and cell classification models. For each institute, models are trained on data from one institute (seen data), with and without data augmentation, and evaluated on test data from the other two institutes (unseen data). The nuclei model is largely unaffected by differences across institutes with an average F1 score of 0.8746 on seen data and 0.8516 on unseen data (Δ −0.0230) when trained with augmentations. In contrast, the nuclei model does not generalise as well when trained without augmentations both within the same institute and on unseen institutes (F1 score 0.8593 vs 0.8242, Δ −0.0351). See Supplementary Table 2 for all results per institute. The cell model, when trained with augmentations, results in a ROC AUC value of 0.9693 on seen data and 0.8410 on unseen data (Δ −0.1283). However, when trained without augmentations, the

0.9722 AUC ROC value on seen data drops to 0.6903 on unseen data (Δ −0.2819), suggesting poor generalisability without data augmentation. See Supplementary Table 3 for all results per institute.

We further investigate the impact of domain shift, where new data is substantially different from trained data, on the generalisability of the cell model. We exclude data from one institute, using a cell model trained on data from the other two, under the assumption that there are no immediately available labelled data. This mimics the practical application of deep learning digital pathology methods where one has access to trained models and new data but no manual labels. In cases where a domain shift from the new data impacts cell classifications, a shift in the cell distributions across one WSI becomes immediately apparent (Supplementary Fig. 6). This is a benefit of assessing a slide using a large number of independent cell classifications against a slide-level biological expectation, which leads to a natively explainable AI approach.

Once a domain shift has been identified, by incorporating a small number of images ($n = 1691$) from the new institute into the training data, the cell model performs equally well across all three institutes (test accuracies of 0.8395, 0.8338, and 0.8757). In practice, as with any application of AI models to new data, one will always have to assess the performance of existing models and the possibility of a domain shift. We include in our released codebase utilities and a walkthrough for generating new nucleus localisation and cell classification datasets for adapting the models to WSIs from new institutes.

## Discussion

We have presented HAPPY, a three-stage hierarchical deep learning pipeline for quantifying cells and tissue microstructures in healthy human placental histology images. Our method localises and classifies 11 cell types and 9 tissue microstructures across a placenta WSI by following an interpretable biological hierarchy and aggregating cellular community information within tissues. The method's cell and tissue outputs match expectations from perinatal pathologists and placental literature and identifies biologically relevant differences between healthy placentas and those with placental infarction. By making tissue predictions at the single-cell level, we can quantify microstructures at a sub-patch resolution, without the need for pixel-perfect manual annotations. We have shown that inductive, message-passing graph neural networks can be applied to cell graphs for node classification built across entire H&E WSIs. To facilitate further development of graph neural network methodologies suitable for cell graphs, we have released the cell graphs and ground truth tissue data from WSIs of two healthy term placentas for the machine learning research community[79].

Additional improvements could be made to ground truth training annotation quality by using immunohistochemical staining of sequential slides to label corresponding H&E cells. The cellular analysis can be extended to include additional cell types such as amniocytes, which are essential for extending this work to placenta membrane roll slides, or to further subcategorise leucocytes and trophoblasts. While applications of HAPPY to other organ histology will require manual annotations of cells and tissue microstructures specific to that organ, we suggest that the present placenta models could serve as domain-relevant pretraining basis for transfer learning. Finally, the HAPPY approach, as implemented here has been applied to H&E specifically but could equally be extended to other high-content imaging domains.

With HAPPY able to quantify cellular and tissue phenotypes in healthy term placenta WSI samples, we present the foundation for this approach to be developed for extracting quantitative metrics of placental health and pathological processes. Despite having been only trained for inference across healthy samples, HAPPY's outputs meaningfully distinguish between healthy samples and those exhibiting a pathology. Training with WSIs from additional data sources and using tissue microstructures found in earlier gestational ages will improve model generalisability, developmental staging, and provide a more comprehensive view of placenta histology. This can become a valuable digital histopathology tool for perinatal pathologists and placenta research. The placenta is an understudied organ and some of the reasons for this is the labour-intensive task of examining placentas, the need for an expert pathologist, and the lack of rich quantitative metrics collected at scale for assessing placenta health.

Already, the HAPPY method can facilitate large-scale morphometric studies of term placenta histology, lending itself to accelerating placenta research and increasing our understanding of the human placenta and its mechanisms.

## Methods

At a high level, HAPPY is structured as a supervised deep learning pipeline of three stages: (i) nucleus localisation, (ii) cell classification and (iii) tissue classification (Fig. 1). First, an image processing module subsections and rescales the WSI into overlapping patches for the nucleus localisation stage. The nucleus localisation stage uses an object detection model to identify the nuclei within each patch. Using these nuclei coordinates, the image processing module crops patches around each nucleus and inputs them to the cell classification stage. The cell classification stage uses an image classification model to classify the nuclei as belonging to one of 11 placental cell types. These cells are input into the tissue classification stage, becoming nodes within a cell graph across the WSI, the edges of which are built to represent probable cellular interactions within the same structure. This cell graph is input into a node classification graph neural network (GNN) which predicts one of 9 tissue microstructure types to which each cell belongs.

### Patient characteristics, slide datasets, histological preparation and ethical approval

The slides used in this work are from a subset of placentas collected as part of routine clinical pathology investigation from three institutes. The first set, from the University of Tartu, Estonia, consists of 110 placentas and 547 slides, collected between 2016-2020 (hereafter referred to as UoT). The second set, from Hadassah Medical Center, Israel, consists of 200 placentas and 831 slides, collected between 2016-2017 (hereafter referred to as HMC). The third set, from Northshore University HealthSystem, University of Chicago, consists of 25 term placentas and 25 slides collected between 2021-2022 (hereafter referred to as NUH). The use of all pseudo-anonymised samples was approved and the requirement for consent was waived by local ethics committees at each institute (UoT approval 289/T-5 by the Research Ethics Committee of the University of Tartu, HMC approval 0735-18-HMO by the Helsinki Committee at the Hadassah Medical Center, NUH exemption of approval EH23-303 by the institutional review board at the Northshore University HealthSystem given that no clinical data was shared beyond selection for histologically normal term placentas). We do not perform sex-based analysis as this information was not available to us in the clinical data provided and sex does not currently inform clinical placental histopathology.

Of the total samples, we used nine parenchyma slides from nine singleton pregnancies from all three institutes of 2nd trimester, pre-term and term samples with both healthy placentas and those exhibiting a pathology for training the nucleus localisation and cellular phenotyping stages. For training the tissue classification stage, we used two parenchyma slides from the placentas of two singleton healthy term pregnancies from UoT and HMC (see Supplementary Table 4 for further information on slides used for training). For inference and whole slide cell and tissue microstructure quantification, we used 30 parenchyma slides from the placentas of 30 singleton healthy term pregnancies from all three institutes and 12 parenchyma slides from the placentas of eight singleton term pregnancies with placental infarction from UoT and HMC (see Supplementary Table 5 for

information on slides used for inference). All slides with placental infarction were reported as clinically significant by perinatal pathologists at respective institutes (K.M and L.S.) and slides were selected such that they contained a region of the infarction on the slide. A second assessment of the healthy slides as histologically normal and those with placental infarction was provided by L.M.E.

Given the heterogeneity and resiliency of the placenta, we select slides from and define 'healthy' as term placentas from pregnancies with no adverse health outcomes during or after pregnancy and where pathology reports and second assessment state that histological sections of the parenchyma were 'normal' and the 'villi correspond to gestational age'. As an example, a portion of these healthy placentas from HMC were from pregnancies which had suspected placenta accreta from a 1st-trimester ultrasound, thereby qualifying them for submission to microscopic examination, but with no resulting complications.

Histology slides were prepared using a standard formalin fixing, paraffin-embedded, Hematoxylin and Eosin (H&E) staining procedure. As per clinical guidelines, appropriate, full thickness sites including both chorionic and basal plates were sampled and 5 µm thickness slices were generated. Slides were scanned and digitised using a Hamamatsu XR, a 3D HISTECH PANNORAMIC 250 Flash III or a Aperio GT 450 scanner at x40 magnification.

## Image processing

The image processing module first selects an appropriate slide reading library, one of libvips[58], openslide[59], or bioformats[60] depending on the slide file format. The slide is partitioned into $1600 \times 1200$ pixel ($177.44 \times 133.08$ µm) patches, with an overlap of 200 (22.18 µm) for the nuclei localisation stage. Patches with mean channel values > 245 or <10 are removed to exclude patches containing no tissue. For the cell classification stage, patches of $200 \times 200$ pixels ($22.18 \times 22.18$ µm) are extracted centred on each nucleus. All patches are extracted and loaded onto devices (CPU or GPU) in memory, without the need for additional on-disk storage to store extracted images. Given that slide scanners output different pixel sizes per micrometre, all patches are rescaled to 0.1109 micrometres per pixel. Metadata and results are stored efficiently in an SQLite database and streamed to hdf5 files with the ability to pause and continue partial inference across a WSI.

## Nucleus localisation

The nucleus localisation stage takes $1600 \times 1200$ ($177.44 \times 133.08$ µm) pixel images extracted from the WSI with a 200 (22.18 µm) pixel overlap to ensure all nuclei are shown whole to the model at least once. Locations with duplicate nuclei predictions within a small radius (4 pixels) generated as a result of this overlap are removed with post-processing. We train a RetinaNet[80] with ResNet-101[81] backbone to predict bounding boxes around nuclei in the image, for which centroid coordinates are saved as the final prediction. The model is first fine-tuned from Coco[82] weights for 40 epochs, with an Adam[83] optimiser, focal loss, and a 0.0001 learning rate with a 0.5 decay every 20 epochs. The model with the highest validation F1 score is then fully trained for 60 epochs with a 0.001 learning rate with the same hyperparameters and the model with the highest validation F1 score is saved. Input images are subject to heavy image augmentation, including various H&E-specific stain augmentations (details of augmentation parameters are described in Supplementary Table 6 and Supplementary Fig. 7).

On the validation and test datasets, model performance is evaluated using the F1 score of identified centroids within a certain distance (<3.3 µm) to the manually labelled ground truth points. This distance is smaller than typical nuclei radii and accounts for minor discrepancies from true centroids in the annotated data. At inference across a WSI using an NVIDIA A100 GPU, the nucleus localisation stage detects ~1000 nuclei per second.

## Cell classification

The cell classification stage takes $200 \times 200$ ($22.18 \times 22.18$ µm) pixel images with each prior predicted nucleus at its centre and classifies these images into one of 11 placental cell types. These include four trophoblast cells: syncytiotrophoblast, cytotrophoblast, syncytial knot, and extravillus trophoblast; five villus mesenchymal-derived cells: fibroblast, Hofbauer cell, vascular endothelial cell, vascular myocyte, and undifferentiated mesenchymal cell; and two non-villus cells: the maternal decidual cell, and leucocyte. The 22.18 µm radius around each nucleus is large enough to capture the entirety of most cell types in addition to contextual information surrounding the cell which may be relevant for prediction (i.e. red blood cells that are near vascular endothelial cells).

We first fine-tune a ResNet-50[81] model from ImageNet[84] weights for 60 epochs, with an Adam[83] optimiser, cross entropy-loss, and a 0.0001 learning rate with a 0.5 decay every 20 epochs. The model with the highest validation accuracy is fully trained for 100 epochs with the same hyperparameters and the model with the highest validation accuracy is saved. Due to class imbalance, minority classes are over-sampled during training to balance the distribution. Images input to the classification model are subject to the same image augmentation as the images used for nuclei detection (Supplementary Table 5). As a final post-processing step, a k-d tree is constructed across all syncytial knot predictions to convert isolated knots into syncytiotrophoblasts and to group clusters of syncytial knot nuclei into a single point. Syncytial knot nuclei which have fewer than 4 neighbours within a 50-pixel radius are relabelled as syncytiotrophoblasts and groups with 4 or more neighbours[85] have their neighbours removed. At inference across a WSI using an NVIDIA A100 GPU, the cell classification stage classifies ~230 cells per second.

## Tissue classification

The tissue classification stage comprises cell graph construction and supervised graph neural network node classification to classify each cell into one of nine placental tissue types. These include the term chorionic villus types: stem villi, anchoring villi, mature intermediate villi, terminal villi, and villus sprouts, the maternal/fetal surfaces: the chorionic plate and basal plate/septum, and areas which in large quantities would indicate pathology[13,65,66,72]: fibrin and avascular villi.

For the cell graph, we define nodes from the outputs of the cellular phenotyping stage, with node features comprising the 64-dimension embedding vectors from the penultimate layer of the cell classifier. The undirected edges connecting the cell nodes are constructed from the intersection of two other edge-building algorithms, k-nearest neighbours (k = 5)[86] and Delaunay Triangulation[87], with the addition of self-loops. This intersection benefits from the more sparsely connected Delaunay Triangulation graph but limits the number of edges which could cross from one tissue boundary to another. This graph construction allows a message-passing model to aggregate cellular information within distinct tissue microstructures while accounting for differences in tissue size and internal cell distances.

We train a randomly initialised, inductive, ClusterGCN[88] model with 16 GraphSAGEConv[89] layers each with 256 hidden units for 2000 epochs with an Adam[83] optimiser, custom weighted cross entropy-loss, a 0.001 learning rate, a batch size of 200 with batch normalisation, and a subgraph sampling size of 400 neighbours. The model with the highest validation accuracy, calculated without neighbourhood sampling and for an intersection graph with k = 8, is saved as the final model. For each node the message-passing algorithm samples and aggregates the node features of nearby nodes up to 16 edge connections away. These aggregated node features are used by the model to predict the tissue type of that node. In this way, the model is aggregating the cellular community which comprises each tissue microstructure to make its prediction. At inference across a WSI using a

laptop CPU, the tissue classification stage classifies ~4500 nodes per second.

A key benefit of the node classification approach is the freedom for the model to assign different tissue types to different sections of the same continuous structure. Placental tissues grow from one another in a tree-like morphology so any cross-sectional cut of what appears to be a single structure may contain multiple valid classifications. For example, a cross section of a mature intermediate villus is likely to have terminal villi branching from it but may also contain fibrin, resulting in three different classes. Additionally, the distinctions between villus types are not necessarily discrete; a terminal villus is distinguished from a mature intermediate villus by having >50% of its stroma taken up by capillaries and by having vasculosyncitial membranes[20,21,38]. However, a section of a mature intermediate villus with unusually many capillaries but no vasculosyncitial membranes, for example, might have nodes that are (mis)classified by the model as a terminal villus section but are not necessarily incorrect.

### Datasets and data annotation

Manual ground truth training, validation and test dataset annotation were performed by C.V. using QuPath[90]. To efficiently train models with a relatively small number of data points (<17k), datasets for the nuclei localisation and cell classification stages were created iteratively by bootstrapping and correcting prior models' predictions in patches of new, unseen slides. Training, validation, and test splits (see Supplementary Table 7) were randomly generated at around a 70/15/15 ratio for each dataset and model performance was evaluated both individually for datasets and in combination.

For the tissue classification stage, ground truth annotations were created by drawing rough boundaries around tissue microstructures, the class of which was assigned to nuclei nodes within the boundary. Validation and test regions of the slide were explicitly chosen such that they were larger than the 16-hop neighbourhood aggregation and contained a similar distribution of tissue microstructures to the training set (See Supplementary Fig. 8). Splitting datasets by region was found to limit information leakage when different cells of same tissues were shared across random dataset splits. This is in contrast to many graph learning datasets which generate dataset splits randomly across nodes.

### Comparison to perinatal pathologists

To assess the accuracy of ground truth training annotations made by C.V. and to judge the relative difficulty of identifying placental tissue types, four practising expert perinatal pathologists, K.M, S.S, W.T.P, L.M.E, performed a similar labelling task across tissue microstructures. Each pathologist was shown a series of images containing a centred tissue microstructure out of 180 total images with the task of labelling the tissue type of that centred structure. Images were generated from a random, class-balanced subset of the ground truth annotations and pathologists were blind to the original annotation and each other's labels. Images were cropped to display some contextual background, similar to the context a 16-layer message-passing GNN may see; however, it was expected that tissue types were identifiable from their cellular composition alone.

Participants were first presented with a Standard Operating Procedure (SOP) and a tutorial (Supplementary Methods 1 and 2). These documents detailed the labelling setup, what data would be collected, how it would be used, and included links to current literature relevant to placental tissue microstructures. Participants were invited to a project in the browser-based software LabelBox[91] where they were sequentially presented with images and could choose one of 12 tissue types for that image. Alternatively, they could state that the type was unclear or not listed and they could leave a comment. Participants were informed that all images came from a healthy term placenta. After completion, their tissue type labels were compared for Cohen's kappa[92] agreement scores against each other, the original annotations, and the model.

### Hardware, software and libraries

All training and inference were performed on a single NVIDIA A100 GPU. Whole slide image processing C libraries were libvips (v8.9.2)[58] using OpenSlide (v3.4.1)[59] or bioformats (v6.11)[60] with libvips taking priority. These were called via their python-bindings pyvips (v2.1.14), openslide-python (v1.2.0) and python-bioformats (v4.0.7). Code was written in Python (v3.10.13) with the PyTorch (v2.0.1)[93] and torchvision (v0.15.2) deep learning framework for the nucleus localisation and cellular phenotyping pipelines and the PyTorch Geometric extension (v2.3.1)[94] for the tissue phenotyping pipeline. Results were recorded in an SQLite database (v3.2.7) and hdf5 files (v3.8.0). WSI visualisation and cell and tissue annotations were done in QuPath (v0.3.1)[90]. Other Python libraries used in analysis and model training included albumentations (v1.3.0), peewee (v3.16.2), pytest (v7.3.1), typer (v0.9.0), visdom (v0.2.4), matplotlib (v3.7.1), pandas (v2.0.1), numpy (v1.24.1), scikit-image (v0.22.0), scikit-learn (v1.3.1), umap-learn (v0.5.3) and seaborn (v0.12.2).

### Reporting summary

Further information on research design is available in the Nature Portfolio Reporting Summary linked to this article.

## Data availability

The datasets generated for training and validating each deep learning model along with trained model weights are available for download at the Google Drive link: https://tinyurl.com/happyplacenta or from Zenodo[95]: 10.5281/zenodo.10535021 with no restrictions. Instructions can be found in the GitHub readme at: https://github.com/Nellaker-group/happy. The two histology slides used for graph model training are available for download under CC BY 4.0 from the BioImage Archive at 10.6019/S-BIAD1045. The remaining in-house placenta histology slides and clinical data are not made available in accordance with existing research ethics committee approvals and data transfer agreements. Pretrained ImageNet weights for the RetinaNet and ResNet-50 models were downloaded in code via PyTorch from https://download.pytorch.org/models/resnet101-5d3b4d8f.pth and https://download.pytorch.org/models/resnet50-0676ba61.pth and will be downloaded programmatically on first model use. Source data are provided with this paper.

## Code availability

Code is available at the following GitHub repository https://github.com/Nellaker-group/happy and at https://doi.org/10.5281/zenodo.10529239[96].

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

## Acknowledgements

C.V. is supported by the EPSRC Center for Doctoral Training in Health Data Science (EP/S02428X/1). C. M. L. is supported by the Li Ka Shing Foundation, NIHR Oxford Biomedical Research Centre, Oxford, NIH (1P50HD104224-01), Gates Foundation (INV-024200), and a Wellcome Trust Investigator Award (221782/Z/20/Z). T.L. was funded by the European Regional Development Fund and the programme Mobilitas Pluss (MOBTP155) and the Estonian Research Council grant (PSG776). The computational aspects of this research were supported by the Wellcome Trust Core Award Grant Number 203141/Z/16/Z and the NIHR Oxford BRC. The views expressed are those of the author(s) and not necessarily those of the NHS, the NIHR or the Department of Health. The Genotype-Tissue Expression (GTEx) Project was supported by the Common Fund of the Office of the Director of the National Institutes of Health, and by NCI, NHGRI, NHLBI, NIDA, NIMH, and NINDS. The data used for supplementary analyses described in this manuscript were obtained from: the GTEx Portal on 24/10/22.

## Author contributions

C.V., C.M.L. and C.N. conceived the study and designed the experiments. C.V. and C.N. wrote the code, C.V. created ground truth annotations and performed the experimental analysis. J.D., V.R., O.D., L.S., K.M., D.H., H.H., and T.L. curated the in-house datasets. S.S., K.M., W.T.P., L.M.E. partook in the pathologist comparison. C.V., C.M.L. and C.N. prepared the manuscript with input, revisions, and approval from J.D., V.R., O.D., S.S., L.S., K.M., W.T.P., D.H., A.F., H.H., T.L., and L.M.E. L.M.E. mentored C.V. on placental biology and perinatal pathology. C.N. and C.M.L. supervised the research.

## Competing interests

The authors declare no competing interests.
