## [Peer Review File · Nature Communications]

Reviewers' Comments:

Reviewer #2:

Remarks to the Author:

This interesting and well written manuscript describes an AI approach to placental histopathology. The model is well described but not in my area of expertise as a perinatal pathologist. Therefore, I will restrict my comments to those involving the placenta.

I think the audience of this journal might not know much about the placenta and so I suggest a figure of a normal placenta (either a cartoon or a gross image) and one or two histology images. I also suggest expanding the introduction a wee bit to talk a little more about the utility of placental pathology to give more umph to the model. I would recommend referencing papers not books.

This one is a good start:

Redline, R. W., D. J. Roberts, M. M. Parast, L. M. Ernst, T. K. Morgan, M. F. Greene, C. Gyamfi-Bannerman, J. M. Louis, E. Maltepe, K. K. Mestan, R. Romero and J. Stone (2022). "Placental pathology is necessary to understand common pregnancy complications and achieve an improved taxonomy of obstetrical disease." Am J Obstet Gynecol.

I would also list the perinatal pathologists by initials whenever they are brought up in the text.

I was confused about the "ground truth" use. Could it be explained a little more in the text?

Of the 11 cell types the model can identify I noticed the omission of amniocytes. Was this studied?

Basically I would just like a little more expansion of the placental part of the manuscript.

Reviewer #3:

Remarks to the Author:

This study introduces a deep learning pipeline, termed 'Histology Analysis Pipeline.PY' (HAPPY), for accurate pathology assessment of the placenta, which is essential for monitoring maternal and newborn health. The placenta's heterogeneity and temporal variability present challenges for histology analysis. HAPPY overcomes these issues by quantifying cell variability and micro-anatomical tissue structures across whole slide images of the placenta in a biological hierarchical pipeline, including nucleus localization, cell classification, and tissue classification. The researchers also provide a set of quantitative metrics from healthy term placentas as a reference for future assessments, i.e. cell proportion and tissue proportion. The study illustrates how these metrics deviate in placentas with clinically significant placental infarction. The authors also showed that the cell and tissue predictions from HAPPY closely align with those from independent clinical experts and placental biology literature, demonstrating its potential as a valuable tool in placenta pathology assessment.

Strengths:

1. The study introduces a novel three-stage deep learning pipeline designed to analyze placenta histology whole slide images (WSIs), offering an advanced framework for pathology assessments.
2. The authors have made the code base publicly available, which greatly encourages reproducibility and further advancements in this area.

Weaknesses:

There are several weaknesses in the experimental design that need to be addressed to justify better the potential clinical usefulness of their results:

1. In the section on Quantitative metrics for placental health (Line 173), the analysis hinges on just 5 parenchyma WSIs of healthy term placentas. This sample size is quite limited, potentially affecting the reliability and generalizability of the provided quantitative reference metrics. The selection process of these 5 WSIs is not clearly explained. It would be beneficial to clarify whether these samples were collected from both centers, and if there are significant differences in cell/tissue proportions between the two institutions.
2. The authors have used data from two different institutes, but they have not presented any analysis on the model's robustness across these centers. To adequately assess the model's

robustness and its generalizability, it would be useful to have the cross-center accuracy of each stage model. For instance, training the model with data from center 1 and evaluating it on data from center 2 could provide insightful results.

Reviewer #4:

Remarks to the Author:

* I am only evaluating the paper from image analysis perspective.

The paper is well-written. The structure is readable and results are presented in an understandable fashion.

I have three main concerns:

- RetiaNet has not been motivated. Why not YOLO or SSD? What specific reason(s) made you choose RetiaNet? A comparison with SOTA in object detection is crucial.
- The unseen data is from the held-out, which means no external validation has been performed. We all understand the challenges of external validation for such a tedious task like cell segmentation. But it is practically impossible to evaluate the performance of deep learning for such sensitive application like placenta health without some notion of external validation.
- The low accuracy of tissue classification (67%) is a concern. The top-2 accuracy of 90% is not comforting unless something like mAP (mean average precision). But why is tissue classification low. What would be the effect if on real unseen data (from other hospitals)?

We would like to thank the reviewers for their suggestions and the editor for inviting a revision of the manuscript. We believe that we have a much improved manuscript thanks to these changes.

To summarise the major changes, we have expanded the placenta parts of the manuscript by adding a new Figure 2 of a placenta parenchyma schematic and extending the introduction, as per Reviewer 2's suggestions. To strengthen our claims of providing quantitative metrics of placental health and to explore model generalisability we have analysed 25 new WSIs from a new institution, as per Reviewer 3 and 4's feedback. We have updated all tables, figures, and reported results to include these new WSIs and we have made sure to keep them unseen to the tissue model. The new section of the Results 'Generalisability and domain shift' explores the effect of institute-bias on the models.

The rest of this document contains point-by-point responses to reviewers.

Reviewer #2:

This interesting and well written manuscript describes an AI approach to placental histopathology. The model is well described but not in my area of expertise as a perinatal pathologist. Therefore, I will restrict my comments to those involving the placenta.

I think the audience of this journal might not know much about the placenta and so I suggest a figure of a normal placenta (either a cartoon or a gross image) and one or two histology images. I also suggest expanding the introduction a wee bit to talk a little more about the utility of placental pathology to give more umph to the model. I would recommend referencing papers not books. This one is a good start:

Redline, R. W., D. J. Roberts, M. M. Parast, L. M. Ernst, T. K. Morgan, M. F. Greene, C. Gyamfi-Bannerman, J. M. Louis, E. Maltepe, K. K. Mestan, R. Romero and J. Stone (2022). "Placental pathology is necessary to understand common pregnancy complications and achieve an improved taxonomy of obstetrical disease." Am J Obstet Gynecol.

We agree that the placenta and the role of placenta pathology was not well explained. We have added Figure 2, a cartoon schematic of a normal term placenta including labelled macro structures (i.e., chorionic villi), an example parenchyma sampling site and a histology image. We have also expanded the first paragraph of the introduction to discuss the utility of placenta pathology.

I would also list the perinatal pathologists by initials whenever they are brought up in the text.

We agree and have now listed the perinatal pathologists whenever they are brought up.

I was confused about the "ground truth" use. Could it be explained a little more in the text?

We agree that this phrasing isn't clear for those outside of the deep learning domain. We have added clarification to all instances where "ground truth" appeared in the text.

Of the 11 cell types the model can identify I noticed the omission of amniocytes. Was this studied?

For now, we have chosen to not include amniocyte detection. It is hard to know where to draw the line when it comes to cell detection (for example, we could have subcategorised leukocytes and further subcategorised the trophoblasts), but for the purposes of quantifying healthy variation within parenchyma slides we believe that the existing cell classes provide a good interpretation of the function of any one placenta. Additionally, in many of our slides the amnion was not present. However, we do have plans to include amniocytes in future work. We have added this omission as a limitation to the Discussion and highlighted it as potential future work, especially in the context of the other placenta slide types.

Basically I would just like a little more expansion of the placental part of the manuscript.

Reviewer #3:

This study introduces a deep learning pipeline, termed 'Histology Analysis Pipeline.PY' (HAPPY), for accurate pathology assessment of the placenta, which is essential for monitoring maternal and newborn health. The placenta's heterogeneity and temporal variability present challenges for histology analysis. HAPPY overcomes these issues by quantifying cell variability and micro-anatomical tissue structures across whole slide images of the placenta in a biological hierarchical pipeline, including nucleus localization, cell classification, and tissue classification. The researchers also provide a set of quantitative metrics from healthy term placentas as a reference for future assessments, i.e. cell proportion and tissue proportion. The study illustrates how these metrics deviate in placentas with clinically significant placental infarction. The authors also showed that the cell and tissue predictions from HAPPY closely align with those from independent clinical experts and placental biology literature, demonstrating its potential as a valuable tool in placenta pathology assessment.

Strengths:

1. The study introduces a novel three-stage deep learning pipeline designed to analyze placenta histology whole slide images (WSIs), offering an advanced framework for pathology assessments.
2. The authors have made the code base publicly available, which greatly encourages reproducibility and further advancements in this area.

Weaknesses:

There are several weaknesses in the experimental design that need to be addressed to justify better the potential clinical usefulness of their results:

1. In the section on Quantitative metrics for placental health (Line 173), the analysis hinges on just 5 parenchyma WSIs of healthy term placentas. This sample size is quite limited, potentially affecting the reliability and generalizability of the provided quantitative reference metrics. The selection process of these 5 WSIs is not clearly explained. It would be beneficial to clarify whether these samples were collected from both centers, and if there are significant differences in cell/tissue proportions between the two institutions.

We agree and we have included the 25 new WSIs to add strength to the reported quantitative reference metrics. All figures, tables, and reported results have been updated with this new data.

To clarify the slide selection process, we have expanded the 'Patient Characteristics, Slide Datasets, and Histological Preparation' section of Methods to specify that all 30 WSIs used for inference are from all three (previously two) centres, with slide-by-slide details in the Supplement.

Regarding whether there are cell/tissue proportion differences between institutions, we do not have the means to objectively assess this, beyond using our methods. In theory, because all slides are from healthy term placentas, there should not be any institute-related bias in the underlying cell and tissue proportions. There may be institute-related bias in the cell and tissue proportions as detected by our method but given the observed variance between slides from the same institution, we believe that we do not currently have the power to detect these differences across institution. However, any noticeable/significant differences between cell and tissue proportions from an institution-bias are more likely to be caused and identified by a domain shift which impacts model performance, as now described in the new section 'Generalisability and domain shift' of the Results.

2. The authors have used data from two different institutes, but they have not presented any analysis on the model's robustness across these centers. To adequately assess the model's robustness and its generalizability, it would be useful to have the cross-center accuracy of each stage model. For instance, training the model with data from center 1 and evaluating it on data from center 2 could provide insightful results.

Thank you, we agree that cross-centre accuracy is an important addition to the manuscript. In the new section of the Results, 'Generalisability and domain shift' we investigate the effect of training the models on one of the now three centres and evaluating against the other two, for each centre. This analysis is also made with and without pixel value altering augmentations, such as our custom stain augmentation, to assess the impact of these augmentations on generalisability.

Reviewer #4:

- I am only evaluating the paper from image analysis perspective.

The paper is well-written. The structure is readable and results are presented in an understandable fashion.

I have three main concerns:

- RetiaNet has not been motivated. Why not YOLO or SSD? What specific reason(s) made you choose RetinaNet? A comparison with SOTA in object detection is crucial.

In part we have continued to use RetinaNet following prior work from our group (arXiv:1804.03270) as we have found it to be a well performing model for nuclei detection in the placenta. To our knowledge, there are no other models which were trained to detect nuclei in the placenta, including mutli-organ models, so we compared our model to the multi-organ nuclei detection of HoVer-Net. HoVer-Net is a popular model for this task and is commonly used as a baseline for challenges (e.g. CoNIC <https://github.com/TissuelmageAnalytics/CoNIC>). Our RetinaNet model gets an F1 score of 0.884 on our test data and HoVer-Net gets an F1 of 0.756 on the CoNSeP dataset and 0.800 on the PanNuke dataset. We have added a comparison to SONNET, which is SOTA on the MoNuSAC dataset with an F1 score of 0.855. While the currently used RetinaNet model is not the current SOTA in object

detection, we have shown that it can achieve comparable SOTA results in our specific domain.

- The unseen data is from the held-out, which means no external validation has been performed. We all understand the challenges of external validation for such a tedious task like cell segmentation. But it is practically impossible to evaluate the performance of deep learning for such sensitive application like placenta health without some notion of external validation.

This is a good point and one we have sought to address with the 25 new WSIs from the new institute. The new section of the Results 'Generalisability and domain shift' explores the effect on model performance when using data from an unseen institute. However, in practice, as with any deep learning and especially so in the digital pathology space, external validation will always be necessary before deploying trained models. We have also added this point to the text and highlighted the utilities our codebase offers for creating new training datasets.

- The low accuracy of tissue classification (67%) is a concern. The top-2 accuracy of 90% is not comforting unless something like mAP (mean average precision). But why is tissue classification low. What would be the effect if on real unseen data (from other hospitals)?

Accuracy in this context is a harsh metric as it is evaluating a kind of sparse segmentation except rather than classifying pixels into tissues, it is classifying cell points into tissues. Points which blur the boundary between two tissue types will have been manually labelled into one class but the model might have a slightly different decision boundary which isn't necessarily incorrect. This isn't reflected in top-1 accuracy but is better captured by top-2 accuracy. Furthermore, the most common tissue types in the placenta are the chorionic villi (chorionic plate, stem villi, mature intermediate villi, terminal villi, and villous sprouts) but, as shown in Figure 6b, rather than being discrete categories they sit along a biological continuum. This is why a high top-2 accuracy better reflects model performance than top-1 accuracy. However, we agree that this was not clearly conveyed in the manuscript, so we have updated the text at 'Evaluation of model performance'.

Additionally, for the purposes of making a slide-level assessment, a top-1 accuracy of 67% from many independent predictions aggregated across a slide is sufficient for capturing the biological signal in a domain with large signal-to-noise ratio (where noise can, for example, come from manual annotation, slide processing and scanning, and/or the biology). The correctly identified signal is shown by how tissue predictions match biological expectation (in 'Quantitative metrics for placental health') and capture differences between healthy slides and those with placental infarction (in 'A Case Study of Placental Infarction').

To address 'the effect if on real unseen data (from other hospitals)', we have made sure to keep the 25 new WSIs from the new institute unseen to the tissue model and incorporated their tissue predictions into the biological validation sections of the manuscript. The predictions on this real unseen data fall within biological expectations for healthy term placentas and as such we believe that the signal is correctly captured by the models in these slides. The differences between healthy term placentas and those with placenta infarction are more pronounced with the addition of this new data.

Reviewers' Comments:

Reviewer #2:

Remarks to the Author:

Thank you for your thoughtful responses to my comments and suggestions.

Reviewer #4:

Remarks to the Author:

All my concerns have been addressed. Thank you.